# FlowLLM: Flow Matching for Material Generation with Large Language Models as Base Distributions

**Anuroop Sriram**
FAIR, Meta
anuroops@meta.com

**Benjamin Kurt Miller**
University of Amsterdam
b.k.miller@uva.nl

**Ricky T. Q. Chen**
FAIR, Meta
rtqichen@meta.com

**Brandon M. Wood**
FAIR, Meta
bmwood@meta.com

## Abstract

Material discovery is a critical area of research with the potential to revolutionize various fields, including carbon capture, renewable energy, and electronics. However, the immense scale of the chemical space makes it challenging to explore all possible materials experimentally. In this paper, we introduce FlowLLM, a novel generative model that combines large language models (LLMs) and Riemannian flow matching (RFM) to design novel crystalline materials. FlowLLM first fine-tunes an LLM to learn an effective base distribution of meta-stable crystals in a text representation. After converting to a graph representation, the RFM model takes samples from the LLM and iteratively refines the coordinates and lattice parameters. Our approach significantly outperforms state-of-the-art methods, increasing the generation rate of stable materials by over three times and increasing the rate for stable, unique, and novel crystals by $\sim 50\%$ – a huge improvement on a difficult problem. Additionally, the crystals generated by FlowLLM are much closer to their relaxed state when compared with another leading model, significantly reducing post-hoc computational cost.

## 1 Introduction

Material discovery holds transformative potential across numerous industries including carbon capture[38], batteries[28], photovoltaics[9], and energy storage[1]. However, the vastness of the chemical space has hindered experimental synthesis of the majority of possible materials. Generative models offer a promising avenue for exploring this untapped potential.

Generating crystalline materials is particularly challenging as it involves simultaneously generating both discrete (atomic types) and continuous values (atomic positions and lattice geometry). While existing approaches, namely autoregressive large language models (LLMs)[11, 6] and denoising models, *e.g.*, denoising diffusion and flow matching [47, 16, 49, 48, 30, 26, 17], have demonstrated success, they exhibit *complementary* strengths and weaknesses. LLMs excel at modeling discrete values, but they can struggle with continuous values due to their reliance on finite precision representations. Conversely, denoising models more effectively handle continuous values and can easily ensure equivariances, but they face challenges with discrete elements.

LLMs also offer the distinct advantage of natural language prompting, enabling versatile and intuitive conditional generation. This capability is further enhanced by training LLMs on vast corpora of chemistry text, equipping them with valuable prior knowledge to generate chemically valid outputs. Queries like "Generate materials with a high bandgap and thermal stability" or "Propose a novel perovskite structure for efficient solar energy conversion" can be directly integrated into the LLM

38th Conference on Neural Information Processing Systems (NeurIPS 2024).

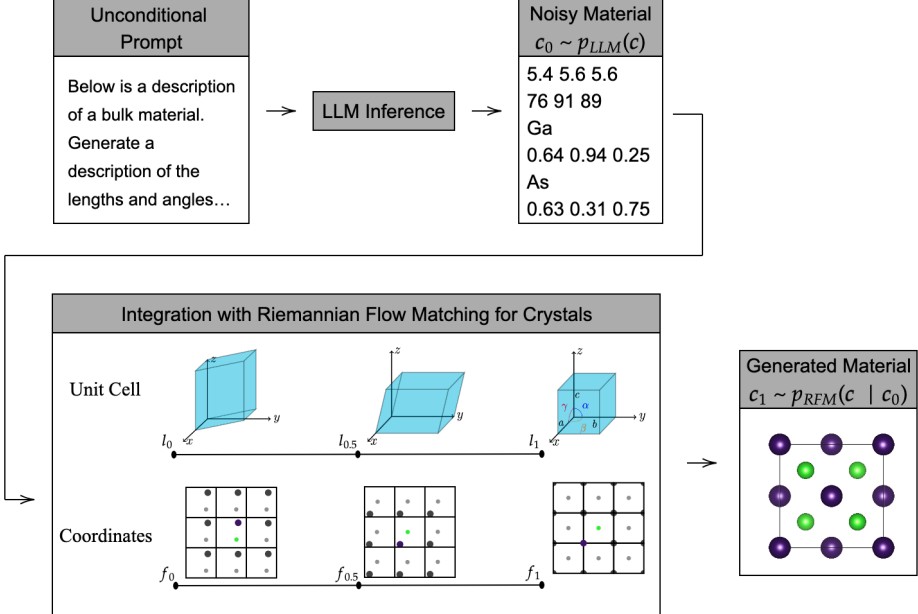

Figure 1: FlowLLM generative process: the fine-tuned LLM is first prompted with an unconditional query to generate an initial material representation. This material is then iteratively transformed by the RFM model to update its atom positions and lattice parameters. The atom types are static in RFM.

prompt, while denoising models typically require bespoke changes to the architecture and training procedure to handle conditional generation.

To harness the strengths of both paradigms, we introduce **FlowLLM**, a novel hybrid approach that uses an LLM to generate an initial material representation, which is iteratively refined with a Riemannian Flow Matching (RFM; [2]) model. This synergistic approach allows us to effectively bridge the gap between discrete and continuous modeling, resulting in a significant improvement in the rate of generation of stable, unique, and novel (S.U.N.) materials. Such materials expand the limited knowledge we have of "material space" and are much more likely to be synthesizable than unstable generations. Our experiments demonstrate that **FlowLLM generates stable materials at over 300% higher rate, and S.U.N. materials at** $\sim 50\%$ **higher rate** compared to prior models, while retaining the LLM's ability to be prompted with natural language instructions.

We offer two interpretations for the effectiveness of our approach. 1) The **LLM learns a good base distribution for RFM**: the LLM's output distribution serves as a learned base distribution for RFM, replacing the common practice of using the uniform base distribution. Since the LLM has been trained on material data, this learned base distribution is closer to the target distribution, greatly simplifying integration with RFM. 2) **RFM refines the output of the LLM**: The LLM generates an approximate material representation due to its finite precision when handling continuous values. The RFM then refines this approximation through iterative denoising, to generate a much more accurate representation.

Our contributions are as follows:

- We introduce FlowLLM, a novel hybrid approach for materials generation that combines LLMs and RFM, effectively leveraging their complementary strengths.
- We demonstrate that FlowLLM significantly outperforms existing state-of-the-art generative models in generating novel and stable materials.
- We show through ablation experiments that our method of combining LLM and RFM models through FlowLLM significantly outperform simpler combination approaches.

Code for training the FlowLLM model is available at https://github.com/facebookresearch/flowmm.

## 2 Related Work

In the past, computational materials discovery relied on generating numerous candidate materials through random atomic substitutions in known materials[42], followed by computationally expensive quantum mechanical screening[19] to assess stability. Genetic algorithms[8, 33], and machine learning models trained to predict energies[37, 25] have accelerated this process, but the fundamental bottleneck of brute force search remains.

Recent research has focused on generative models that directly produce stable materials, bypassing brute-force search. Diffusion models, either combined with Variational Autoencoders (VAEs) for partial variable prediction[47] or jointly diffusing all variables[16, 48, 49] have shown promise. Additionally, Riemannian Flow Matching[26], Normalizing Flows [45], and Variational Autoencoders[34] have also been adapted for material generation.

A parallel line of work utilizes autoregressive Large Language Models (LLMs) for material generation [6, 11], representing materials as a sequence of discretized tokens. Pretraining these models on natural language imbues them with powerful prior knowledge not attainable by other approaches.

## 3 Preliminaries

Our approach models probability distributions over crystal lattices, defined as periodic arrangements of atoms in three-dimensional space. A crystal lattice is created by tiling a fundamental unit cell, where the unit cell contains a specific atomic configuration, forming the entire lattice when repeated. In this section, we present a high-level overview of crystal representations, building up to explain our model in section 4. Background details for the crystal representation are in appendix A.

**Crystal representation**   In the paper, we represent an $n \in \mathbb{N}$ atom crystal in a product space: $c := (a, f, l) \in \mathcal{C}$, indicating the atom types, positions and unit cell geometry, respectively [47, 26]. The atom types are represented by a matrix of categorical vectors: $a := \left[a^1, \ldots, a^n\right]$, where $a^i \in \mathcal{A}$. The atomic coordinates are represented using fractional coordinates within the unit cell, $f := \left[f^1, \ldots, f^n\right]$, where $f^i \in \mathcal{F} = \mathbb{T}^3$ with $\mathbb{T}$ denoting the unitary length, flat torus manifold, *i.e.*, the fractional coordinates satisfy periodic boundary conditions; that is, the atoms "wrap around" the unit cell. The unit cell geometry is defined using lattice parameters $l \in \mathcal{L}$, where $\mathcal{L}$ is the space formed by a 6-tuple of three side lengths $(a, b, c) \in \mathbb{R}^+$ (Å, i.e. Angstrom) and three internal angles $(\alpha, \beta, \gamma) \in [60°, 120°]$. This representation is not unique as the same crystal can be produced by different choices of unit cell. To make the representation unique, we select the minimum-volume unit cell and employ Niggli reduction [10] that uniquely determines the unit cell parameters.

**Equivariance & Invariance**   Given a group $G$ with $g\cdot$ denoting a group action for some $g \in G$, a function $f : \mathcal{X} \to \mathcal{Y}$ is called $G$-*equivariant* if $\forall x \in \mathcal{X}, \forall g \in G, f(g \cdot x) = g \cdot f(x)$, while it is called $G$-*invariant* if $\forall x \in \mathcal{X}, \forall g \in G, f(g \cdot x) = f(x)$. Since a crystal is not uniquely defined by any particular representation $c$ but an infinite set, we know that the data distribution has a $G$-invariant density, where $G$ represents symmetries of a crystal.

**Symmetries of crystals**   Concretely, our crystal representation exhibits multiple symmetries that we detail here. The symmetric group $S_n$ on $n$ atoms permutes the atom indices: $\sigma \cdot c = \left(\left[a^{\sigma(1)}, \ldots, a^{\sigma(n)}\right], \left[f^{\sigma(1)}, \ldots, f^{\sigma(n)}\right], l\right)$. The special Euclidean group SE(3) consists of orientation preserving rigid rotations and translations: $(Q, T)$ where $Q \in \mathrm{SO}(3)$ and $T \in [-\frac{1}{2}, \frac{1}{2}]^{3\times 1}$ denote 3D rotations and translations respectively. This element transforms the crystal as: $(Q, T) \cdot c = (a, f + \mathcal{T}\mathbf{1} - \lfloor f + \mathcal{T}\mathbf{1} \rfloor, l)$. We emphasize that the representation $c$ is completely invariant w.r.t. $Q$ because lattice parameters do not contain orientation information. Since these represent symmetries fundamental to crystals, the data distribution $q(c)$ is invariant to these group operations.

## 4 Method

Our goal is to fit a parametric generative model $p(c; \theta)$ to approximate the distribution of known meta-stable materials $q(c)$ using a dataset of samples. The distributions $p$ and $q$ are defined on the Riemannian manifold $\mathcal{C}$. Our FlowLLM model generates samples from the parametric distribution

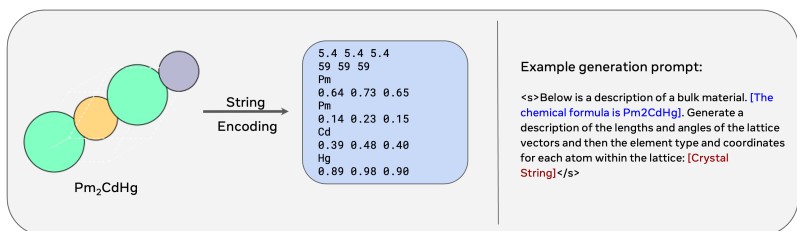

Figure 2: Left: String encoding of materials used to train the LLM based on Gruver et al.[11]. Right: An example prompt used during training. The conditioning information in blue is optional, and can be replaced with conditioning on other properties as well. The text in red is replaced with the crystal string representation shown on the left.

using a two-step procedure (see figure 1). First it samples the LLM, then it refines the LLM output using RFM, like so:

$$c_0 \sim p_{\text{LLM}}(c; \theta_0), \tag{1}$$

$$c_1 \sim p_{\text{RFM}}(c|c_0; \theta_1) \tag{2}$$

where $p_{\text{LLM}}$ is modeled using a large language model[6, 11], and $p_{\text{RFM}}$ is modeled using Riemannian Flow Matching (RFM)[3, 26], and $\theta = (\theta_0, \theta_1)$. Both the LLM and RFM frameworks are trained to estimate the data distribution over meta-stable crystals on samples from the Materials Project [15].

**Overview of training**   First, we fine-tune an LLM to generate string representations of meta-stable materials [11]. Once trained, we can sample the LLM distribution using next token prediction, optionally conditioning on a prompt (see figure 2). Next, we train the RFM model using the FlowMM objective [26] where, conditioned on the chemical formula, will learn to transport between the LLM's model distribution and the data distribution. The full training process is described in Algorithm 1.

**Overview of sampling**   We give the standard prompt to the LLM and allow it to do next token prediction until it produces a stop token. As long as all atom types are actual elements and the lattice parameters are physical, we move forward. Otherwise we reject the sample. Then, we convert the text to a crystal representation that serves as the initial sample. This sample's fractional coordinates $f$ and lattice parameters $l$ are iteratively refined by the RFM model to produce the final sample of FlowLLM. This sampling process is illustrated in figure 1.

### 4.1   Large Language Model ($p_{\text{LLM}}$) for Crystals

LLMs define a distribution over sequences through an autoregressive decomposition, $\prod_{t=1}^{T} p(w_{t+1}|w_{0:t})$, where each $p(w_{t+1}|w_{0:t})$ follows a categorical distribution conditioned on all previous tokens ($w_{0:t}$) in the sequence. Our LLM model closely follows Gruver et al. [11].

**Tokenization**   Language models interact with strings in text datasets after the string is converted into a sequence of tokens. The choice of tokenizer can have a large impact on the performance of the language model. In terms of tokens, we represent a crystal $c$ using fixed precision numbers – two decimal places for fractional coordinates, and one for lattice lengths. Angles are represented as integers. Atom types are represented as discrete tokens. We use LLaMA-2 models [41] for our LLM architecture since these models break numbers into a sequence of digits, which has been shown to dramatically improve performance on arithmetic tasks [23].

**Training**   We rely on the extensive pretraining of LLaMA-2 models to instill useful biases over numerical operations. To train $p_{\text{LLM}}$, we fine-tune a pre-trained LLaMA-2 model on a dataset of crystal structures represented as strings along with a prompt indicating that the model should generate bulk materials by writing the lattice in lengths and angles along with atom types and coordinates. An example of such a representation along with a prompt is shown in figure 2.

The flexibility of LLMs allows us to optionally include different kinds of conditional information in the prompt such as the chemical formula. We can also solve other tasks such as infilling by making changes to the prompt. For this hypothetical conditional generation, the prompt could include a desired

---

**Algorithm 1** FlowLLM training

---

1: **Input:** Training dataset of materials: $\mathcal{D} = \{c^i\}$, Pre-trained LLM: $p_{\text{LLM}}$, RFM velocity network: $v_t^{\theta_1}$, Number of RFM training samples: $N_{tr}$.

**// Step 1: Fine-tune the LLM**
2: Fine-tune $p_{\text{LLM}}$ on $\mathcal{D}$ following the procedure from Gruver et al.[12]

**// Step 2: Sample the LLM to generate training data for the RFM model**
3: Initialize $\tilde{\mathcal{D}} \leftarrow \varnothing$
4: **for** i = 1: $N_{tr}$ **do**
5:     Sample $c_1^i \sim \mathcal{D}$ with replacement
6:     Sample $c_0^i \sim p_{\text{LLM}}(\cdot|\theta_0)$ using a prompt conditioned on the formula of $c_1^i$
7:     $\tilde{\mathcal{D}} = \tilde{\mathcal{D}} \cup \{(c_0^i, c_1^i)\}$
8: **end for**

**// Step 3: Train the RFM model on $\tilde{\mathcal{D}}$**
9: **while** not converged **do**
10:     Sample $(c_0, c_1) \sim \tilde{\mathcal{D}}$, $t \sim \mathcal{U}([0, 1])$
11:     $c_t := \exp_{c_0}(t \log_{c_0}(c_1))$
12:     $\mathfrak{L}(\theta_1) = \|v_t^{\theta_1}(c_t) - u_t(c_t|c_1)\|^2$
13:     Take gradient descent step on $\nabla_{\theta_1} \mathfrak{L}(\theta_1)$
14: **end while**

---

chemical formula, material properties, or a combination of such information. In this work, we used the same conditioning used in Gruver et al.[11], and we leave a more detailed study of this to future work.

**Sampling** To generate sequences from the model, the conditional distribution is sampled sequentially. The sampling procedure is modulated to control the diversity and sampling speed using the temperature ($\tau$) and nucleus size ($P$) hyperparameters of nucleus sampling [13]. Temperature controls the entropy of the conditional distributions, introducing a trade-off between diversity and mode sampling. The nucleus size limits the number of tokens that can be sampled. Given a nucleus size $P$ with $0 < P \leq 1$, sampling is restricted to the most probable tokens with cumulative probability $P$.

**Symmetries in LLMs** The LLM architecture does not inherently produce a symmetric density, *i.e.*, the distribution of meta-stable crystals that the LLM learns is *not* symmetric according to the fundamental properties of crystals. We perform no fractional coordinate data augmentation via translation, and no token permutation data augmentation. Unlike the other symmetries, rotation invariance holds for the learned LLM distribution due to our choice of representing the unit cell with lattice parameters.

### 4.2 Riemannian Flow Matching ($p_{\text{RFM}}$) for Crystals

**Riemannian Flow Matching** RFM produces a Continuous Normalizing Flow [3], *i.e.*, a continuous, parametric, diffeomorphism between the LLM base distribution $p_0 := p_{\text{LLM}}$ and an approximation to our target distribution $p_1 \approx q$. To model $p_{\text{RFM}} := p_1$, we fit a time-dependent vector field $v_t^{\theta_1}$ that has been adapted to crystals and is implemented using a neural network with parameters $\theta_1$. Continuous Normalizing Flows are computationally expensive to train using maximum likelihood, but an alternative objective called Conditional Flow Matching [22] is more stable and scales better. The objective was generalized to Riemannian manifolds [2], and specifically to labeled point clouds with periodic boundary conditions, *i.e.* crystals, by Miller et al. [26].

Concretely, each point $c \in \mathcal{C}$ has an associated *tangent space* $\mathcal{T}_c \mathcal{C}$ with an inner product $\langle u, v \rangle$ for $u, v \in \mathcal{T}_c \mathcal{C}$, enabling the definition of distances, volumes, angles, and minimum length curves (*geodesics*). The geodesics for any $\mathcal{C}$ that we consider can be written in closed form using the exponential and logarithmic maps. The geodesic connecting $c_0, c_1 \in \mathcal{C}$ at time $t \in [0, 1]$ is

$$c_t := \exp_{c_0}(t \log_{c_0}(c_1)), \tag{3}$$

where $\exp_\square$ and $\log_\square$ are the exponential and logarithm maps for the manifold $\mathcal{C}$. These geodesics help define the supervision signal used to train RFM.

Our RFM generative model $v_t^{\theta_1} : [0,1] \times \mathcal{C} \to \mathcal{TC}$ is parameterized as a time-dependent, smooth vector field. Training proceeds by regressing onto conditional vector fields $u_t(\boldsymbol{c}|\boldsymbol{c}_1)$ that generate single data points $\boldsymbol{c}_1$. For the geodesic path, this corresponds to $u_t(\boldsymbol{c}|\boldsymbol{c}_1) = -\frac{1}{1-t}\log_{\boldsymbol{c}_1}(\boldsymbol{c})$. The general RFM training objective is then:

$$\mathfrak{L}(\theta_1) = \mathbb{E}_{t,p(\boldsymbol{c}_0)q(\boldsymbol{c}_1)}\|v_t^{\theta_1}(\boldsymbol{c}_t) - u_t(\boldsymbol{c}_t|\boldsymbol{c}_1)\|^2. \tag{4}$$

Since we only use flat manifolds, $\|\cdot\|$ is the Euclidean norm. At the optimal solution, $v_t^{\theta_1}$ generates $p_t$ with endpoints $p_0 = p$, $p_1 = q$. At sampling time, we draw a sample from $\boldsymbol{c}_0 \sim p$ and solve the ordinary differential equation $\frac{d}{dt}\boldsymbol{c}_t = v_t^{\theta_1}(\boldsymbol{c}_t)$ with initial value $\boldsymbol{c}_0$ at $t = 0$; the solution at $t = 1$ is then the sample from our RFM model.

**Geometry of $\mathcal{F}$** We apply the conditional vector field for a point cloud living on a $n \times 3$-dimensional product of flat tori invariant to global translations, *i.e.* fractional coordinates with periodic boundary conditions [26]. This is a geodesic path, which may cross the periodic boundary:

$$\exp_{f^i}(\dot{f}^i) := f^i + \dot{f}^i - \lfloor f^i + \dot{f}^i \rfloor, \quad \log_{f_0^i}(f_1^i) := \frac{1}{2\pi}\operatorname{atan2}\left[\sin(\omega^i), \cos(\omega^i)\right], \tag{5}$$

where $\omega^i := 2\pi(f_1^i - f_0^i)$, and $\dot{f}^i \in \mathcal{T}_{f^i}\mathcal{F}^i$ for $i = 1, \ldots, n$. Computing the geodesic of $n$ atoms amounts to an atom-wise application of $\log_{\boldsymbol{f}_0}$ on $\boldsymbol{f}_1$ and $\exp_{\boldsymbol{f}}$ on $\dot{\boldsymbol{f}} \in \mathcal{T}_{\boldsymbol{f}}\mathcal{F}$ respectively. Additionally, following Miller et al. [26] we address translation-invariance by removing the mean torus translation:

$$u_t^{\mathcal{F}}(\boldsymbol{f} \mid \boldsymbol{f}_1) := \log_{\boldsymbol{f}_1}(\boldsymbol{f}) - \frac{1}{n}\sum_{i=1}^{n}\log_{f_1^i}(f^i). \tag{6}$$

**Geometry of $\mathcal{L}$** The space of lattice parameters, $\mathcal{L} := \mathbb{R}^{+3} \times [60, 120]^3$, is a Euclidean space with boundaries. We can ignore these boundaries for the lattice lengths in $\mathbb{R}^{+3}$ since (i) the data does not lie on the boundary ($a, b, c > 0$) and (ii) we can clamp our base distribution to be positive with rejection. The boundary issue for the lattice angles $\alpha, \beta, \gamma$ can be addressed [26] using a diffeomorphism $\varphi \colon [60°, 120°] \to \mathbb{R}$ to *unconstrained space*, applied element-wise to each angle:

$$\varphi(\eta) := \operatorname{logit}\left(\frac{\eta - 60}{120}\right), \quad \varphi^{-1}(\eta') = 120\,\sigma\,(\eta') + 60, \tag{7}$$

where $\sigma(.)$ and logit are the sigmoid and the log-odds functions, respectively. We directly apply RFM in the unconstrained space, and for sampling, we map the angles back into $[60°, 120°]$ using $\varphi^{-1}$.

**The RFM training objective** With this formulation, our training objective based on (4) becomes:

$$\mathbb{E}_{t,p_{\text{LLM}}(\boldsymbol{f}_0,\boldsymbol{l}_0|\boldsymbol{a})q(\boldsymbol{f}_1,\boldsymbol{l}_1,\boldsymbol{a})}\left[\frac{\lambda_{\boldsymbol{f}}}{3n}\left\|v_t^{\mathcal{F},\theta_1}(\boldsymbol{c}_t) + \log_{\boldsymbol{f}_1}(\boldsymbol{f}_0) - \frac{1}{n}\sum_{i=1}^{n}\log_{f_1^i}(f_0^i)\right\|^2 \right. \tag{8}$$

$$\left. + \frac{\lambda_{\boldsymbol{l}}}{6}\left\|v_t^{\mathcal{L},\theta_1}(\boldsymbol{c}_t) + \boldsymbol{l}_0 - \boldsymbol{l}_1\right\|^2\right],$$

where we now use $p_{\text{LLM}}$ as the base distribution, and $\boldsymbol{c}_t = (\boldsymbol{f}_t, \boldsymbol{l}_t, \boldsymbol{a})$. The loss coefficients $\lambda_{\boldsymbol{f}}, \lambda_{\boldsymbol{l}} \in \mathbb{R}^+$ are hyperparameters. We use a graph neural network (GNN) inspired by [36, 16, 26] for $v_t^{\theta_1}(\boldsymbol{c})$. This GNN enforces equivariance to atom permutations via message passing, invariance to atom translation by featurizing graph edges as relative displacements of nodes, and invariance to rotations by our choice of lattice representation. See appendix B for more details about the GNN architecture.

## 4.3 Consequences of using an LLM as the base distribution

**Model symmetries** Just like the LLM, the orientation-invariant representation of the unit cell leads to global rotation invariance. However, permutation and translation symmetries are not so simple. If the parameterization of the RFM velocity field is $G$-equivariant, and the *base distribution is $G$-invariant*, then the model density is $G$-invariant [18]. We use graph neural networks [36, 40, 27, 44, 7, 21, 31, 51], and additional projections [26], to ensure that the RFM velocity predictions are $G$-equivariant to both permutation and translation. However, we will generally *not* recover a translation

invariant density because the base distribution defined by the LLM is *not* invariant to translation. The density *will be* permutation invariant in our RFM representation because the each atom is a node in an unordered point cloud and the LLM ordering is ignored by the RFM, but the density *will not be* permutation invariant in the text representation, due to the LLM's lack of token permutation invariance.

Empirically, we do not find the lack of exact invariance to be a problem, and FlowLLM outperforms methods with exact invariance (section 5). This is because an LLM trained to generate crystals is approximately invariant to crystal symmetries. This was verified by Gruver et al.[11] who proposed a new metric, *Increase in Perplexity under Transformation (IPT)*, to quantify this approximation:

$$\text{IPT}(s) = \mathbb{E}_{g \in G}[\text{PPL}(t_g(s)) - \text{PPL}(t_{g^*}(s))] \tag{9}$$

where $g^* = \arg \min \text{PPL}(t_{g^*}(s))$, and PPL is the perplexity of the sequence, the exponent of the length-normalized cross entropy loss, $\text{PPL}(s) = 2^{\text{CE}(s)/n}$. They find that a well-trained LLM obtains a small IPT value, implying that it is approximately invariant.

**Invalid crystals**  The LLM base distribution is not constrained to $\mathcal{C}$, *i.e.* the LLM can generate invalid crystals. We find that this is extremely rare and easy to detect. In such cases, we simply reject that sample, and draw a new sample until we get a valid crystal. Empirically, we found this rejection rate to be $\sim 0.5\%$ with a softmax temperature of 0.7.

**Text is not continuous in $\mathcal{L}$ or $\mathcal{F}$**  The LLM base distribution only takes non-zero values over a small number of discrete points due to the use of finite precision representations. For example, we represent fractional coordinates with only 2 decimal places, so they can only take one of 100 distinct values. We can mitigate this problem by adding a small amount of random zero-mean gaussian noise to all continuous values predicted by the LLM. Empirically, we do not observe any noticeable difference in performance due to this added noise (see appendix F).

## 5 Experiments

### 5.1 Setup

We trained our model on the widely used MP-20 dataset[1] of inorganic crystalline materials[47]. MP-20 comprises 45,231 materials, a subset of the Materials Project[15] containing up to 20 atoms known to be metastable (see section 5.2).

We first train our LLM independently using the various prompting strategies described in Section 4. Unless otherwise specified, we employed a pretrained LLaMA-2 70B model [41] for all experiments, that was fine-tuned with the Low-Rank Adapters (LoRA) method [14] using PyTorch[32] and Transformers[46].

Next, we trained the RFM model using the fine-tuned LLM (with frozen weights) as the base distribution and the MP-20 dataset as the target distribution. For computational efficiency, we sampled a large number ($N_{tr}$) of examples from the base distribution in advance, and used the same set for all of our training runs. To create this set, we sampled $N_{tr}$ materials, with replacement from MP-20, and queried the LLM with a prompt conditioned on the chemical formula of each of these materials. This results in a set of $N_{tr}$ pairs, $\{(\mathbf{c}_0^i, \mathbf{c}_1^i)\}_{i=0}^{N_{tr}}$, of LLM generated materials and ground truth materials that constitutes the training set for the RFM model. We list the hyperparameter values used in our experiments in appendix C.

To generate new samples, we first generate a material from the LLM using an unconditional query. We then perform an integration with the RFM model, starting from this LLM-generated material. During sampling, we can adjust hyperparameters such as temperature $\tau$, nucleus probability $P$, and the number of integration steps to achieve different trade-offs between diversity, accuracy, and efficiency.

### 5.2 Metrics

Our primary metrics are *Stability Rate*, the percentage of generated materials that are thermodynamically stable, a key indicator of synthesizability, and the *S.U.N. rate*, the percentage of materials that

---

[1]Publicly available at `https://github.com/txie-93/cdvae/tree/main/data/mp_20`

| Method | LLM Params | Integ. Steps | Validity (%) ↑ | | Coverage (%) ↑ | | Property ↓ | | Stability Rate (%) ↑ | SUN Rate(%) ↑ |
|---|---|---|---|---|---|---|---|---|---|---|
| | | | Structural | Composition | Recall | Precision | wdist ($\rho$) | wdist ($N_{el}$) | MP-2023 | |
| CDVAE [47] | – | 5000 | **100.00** | 86.70 | 99.15 | 99.49 | 0.688 | 0.278 | 1.57 | – |
| DiffCSP [16] | – | 1000 | **100.00** | 83.25 | **99.71** | 99.76 | 0.350 | 0.125 | 5.06 | 3.34 |
| FlowMM [26] | – | 1000 | 96.85 | 83.19 | 99.49 | 99.58 | **0.239** | **0.083** | 4.65 | 2.34 |
| CrystalLLM (70B) [11] | $\tau = 0.7$ | – | 99.6 | **95.4** | 85.8 | 98.9 | 0.81 | 0.44 | 5.28 | – |
| FlowLLM-Types | $\tau = 0.5, P = 0.9$ | 750 | 99.96 | 93.32 | 96.85 | 99.78 | 0.846 | 0.209 | 8.79 | – |
| | $\tau = 0.9, P = 0.9$ | 750 | 99.88 | 91.69 | 97.18 | 99.76 | 1.14 | 0.20 | 8.95 | – |
| FlowLLM | $\tau = 1.0, P = 0.9$ | 250 | 99.81 | 89.05 | 99.06 | 99.68 | 0.66 | **0.09** | 10.07 | 4.89 |
| | $\tau = 0.7, P = 1.0$ | 250 | 99.88 | 89.45 | 99.06 | 99.71 | 0.73 | 0.14 | 13.03 | 4.88 |
| | $\tau = 0.7, P = 0.9$ | 250 | 99.94 | 90.84 | 96.95 | **99.82** | 1.14 | 0.15 | **17.82** | **4.92** |

Table 1: Results for material generation on the MP-20 dataset. Stability rate is the percentage of generated materials with $E^{\text{hull}} < 0.0$ & $N$-ary $\geq 2$.

are stable, unique and novel. Since computing stability is computationally expense, Xie et al. [47] proposed a number of proxy metrics. We explain these metrics in more detail in appendix D.

One key difference in evaluation between the proxy metrics and the stability metrics is the use of pre-relaxation and relaxation techniques. Proxy metrics are computed on raw samples without any further processing. Stability metrics are computed on structures that are first pre-relaxed using CHGNet[5] then relaxed using Density Functional Theory.

Density Functional Theory is extremely expensive, even with speedups using pseudo-potentials[20]. Ideally, the generative model can generate many S.U.N. structures that are already close to their relaxed ground state. Generating structures close to ground state may also indicate that the model has done a better job capturing the data distribution. It can also speed up or obviate the need for relaxing the generated structures, which has huge computational benefits. We include several additional metrics to measure the closeness of generated and corresponding ground state structures, that are described in appendix E.

## 5.3 Results

We compare our model to four prior methods: CD-VAE[47], a hybrid Variational Autoencoder & diffusion model; DiffCSP[16], a diffusion model; FlowMM[26], a Riemannian Flow Matching model; and CrystalLLM[11], which fine-tunes a LLaMA-2 model on materials represented as sequences. The LLM and RFM components of FlowLLM closely resemble the formulations in CrystalLLM and FlowMM , respectively. To compare different models, we generate 10,000 new structures from each model and compare the metrics described in section 5.2.

Our main results are presented in table 1. On the most important metrics, namely the Stability & S.U.N. rates, FlowLLM significantly outperforms all prior methods across various LLM sampling parameters. For our best FlowLLM model ($\tau = 0.7, P = 0.9$), $17.82\%$ of the generated structures are stable, out of which $48\%$ are novel (not similar to any training or validation structure). Of the remaining structures, $58\%$ are unique, leading a to a S.U.N. rate of $4.92\%$. **FlowLLM obtains a $\sim 300\%$ higher stability rate and $\sim 50\%$ higher S.U.N. rate than the best prior model!**

Figure 3a shows histograms comparing the $E^{\text{hull}}$ values of generated materials from FlowLLM compared to prior models. Clearly, FlowLLM generates many more materials with lower $E^{\text{hull}}$ values than the other models.

The results on proxy metrics, on the other hand, remain mixed. Diffusion and flow matching methods excel on Coverage Recall, while CrystalLLM has the best Composition Validity. FlowLLM achieves the best compromise between coverage and validity, potentially explaining its superior Stability & S.U.N. rates. It is important to note that many of these metrics have become saturated, offering limited discriminatory power for evaluating state-of-the-art models. As a result, we anticipate a decreased reliance on these metrics in future research.

**Comparison of generated and relaxed structures** While the stability rate and S.U.N metrics capture whether the generated structures can be relaxed to stable / S.U.N. states, they do not address the question: *How close are the generated structures to their relaxed state?* To answer this question, we compared generated structures to those same generated structures after relaxation using CHGNet, computing the following metrics between generated and CHGNet relaxed states: *Match Rate* and *RMSD*, as defined by `StructureMatcher`, along with the $\Delta$-*Energy* and the average *Num steps* between the states. Definitions for these metrics can be found in appendix E.

| Method | Match Rate (%) ↑ | RMSD (Å) ↓ | $\Delta$-Energy (eV/atom) ↓ | Num Steps ↓ |
|---|---|---|---|---|
| FlowMM | 74.3 | 0.096 | 0.3031 | 191.98 |
| FlowLLM | **94.9** | **0.023** | **0.0898** | **37.97** |

Table 2: Comparison of generated and corresponding ground state structures from the CHGNet relaxation. Compared to FlowMM, FlowLLM generates structures much closer to the ground state.

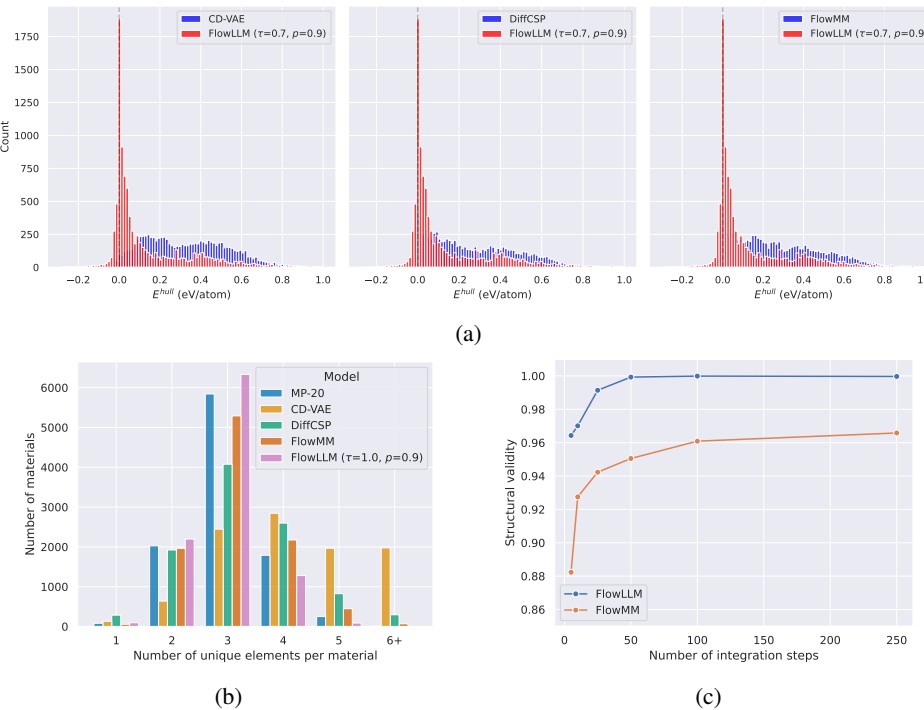

(a)

(b)                                    (c)

Figure 3: (a) Histogram of $E^{\text{hull}}$ values comparing FlowLLM with prior models. The dashed line shows thermodynamic stability threshold ($E^{\text{hull}} = 0$). (b) Histogram of N-ary compared to the data distribution. (c) Structural validity as a function of number of integration steps.

Table 2 shows a comparison of FlowMM and FlowLLM. The samples generated by FlowLLM are significantly closer to ground state compared to FlowMM, according to our metrics.

**Importance of learned base distribution**  One motivation for a hybrid LLM-RFM model is to leverage the LLM's superior ability to generate accurate atom types compared to denoising models. To isolate this effect, we trained the *FlowLLM-Types* model, following a similar procedure as FlowLLM but using simple base distributions for lattice parameters and fractional coordinates identical to those used in FlowMM[26]. Thus, the LLM only contributes to atom type prediction in this model. Despite this simplification, FlowLLM-Types still surpasses prior models on the Stability Rate metric (table 1), highlighting the benefits of employing an LLM for atom type prediction. The stability rate of FlowLLM-Types remains considerably lower than that of FlowLLM, underscoring the substantial value of using learned base distributions.

**N-ary analysis**  The number of distinct element types in a material is called the *N-ary* value of that material. Figure 3b compares the distribution of N-ary values for different models with the target data distribution. FlowMM and FlowLLM  match the data distribution better than the diffusion models, which tend to generate too many materials with high n-ary.

**Number of RFM integration steps**  Compared to diffusion and flow matching models which require hundreds or thousands of integration steps, FlowLLM is able to converge in as little as 50 steps (figure 3c). This is not surprising given our use of a learned base distribution.

# 6 Discussion

The discovery of novel, stable materials holds the potential to help revolutionize numerous industries, but progress has been slow due to the high computational costs involved. Widely used random structure search methods[33] yield less than a 1% success rate in identifying stable materials. Given the substantial cost of validating generated structures using density functional theory, improving this rate is of paramount importance.

Recent breakthroughs with denoising models[16, 26] and large language models[11] have increased the stability rate to $\sim 5\%$, a significant improvement over traditional approaches. In this work, we propose a novel generative model which harnesses the strengths of both paradigms to further increase this number by over $3\times$, representing a major advancement in the field.

**Limitations** While FlowLLM excels at generating stable materials, a key limitation is its lack of end-to-end differentiability. This hinders its direct application to inverse design, where generative models are optimized to generate material with specific properties, as explored in prior work using denoising models[49, 47]. Future research could investigate extending FlowLLM for inverse design.

**Broader impact** This work can accelerate the discovery of new materials for renewable energy, electronics, and carbon capture, ultimately benefiting society by enabling more efficient and sustainable technologies. However, the adoption of generative models also raises concerns, such as the creation of harmful substances and access inequalities.

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

## A  Crystal Representations Details

**Atomic types**   The representation of atomic number is dependent on the model processing the data. In the LLM, the name of the element can be written into the text representation directly. This can be a string or single token, depending on LLaMA-2's tokenization. In the RFM framework, we applied a one-hot representation.

**Unit cell geometry**   Throughout the paper and in our implementation, we represent the unit cell using lengths and angles; however, there is another representation relevant for defining the fractional coordinates and better expressing crystal symmetries. The unit cell can be defined by a matrix of Cartesian column vectors $\tilde{l} := \left[\tilde{l}^1, \tilde{l}^2, \tilde{l}^3\right] \in \tilde{\mathcal{L}} = \mathbb{R}^{3\times3}$. This representation has strictly more information than $l$, since it also defines the orientation of the unit cell. This orientation is irrelevant in our paper, since we want rotation invariance. That's why we choose $l$ in the first place.

**Fractional coordinates**   Now that we have the representation $\tilde{l}$, we can define fractional coordinates. Recall, atomic positions are typically represented using Cartesian coordinates $\boldsymbol{x} := \left[x^1, \ldots, x^n\right] \in \mathcal{X} = \mathbb{R}^{3\times n}$ with coordinates in the rows and atoms in the columns. The Fractional coordinate representation is defined $\boldsymbol{f} := \tilde{l}^{-1}\boldsymbol{x} = \left[f^1, \ldots, f^n\right] \in \mathcal{F} = [0, 1)^{3\times n}$.

## B  Graph Neural network in the RFM Model

In this section, we describe the graph neural network used in our RFM model. Our GNN model is inspired by the GNNs used in FlowMM[26] and DiffCSP[16], which in turn adapted the EGNN [36] model for fractional coordinates,

$$\boldsymbol{h}^i_{(0)} = \phi_{\boldsymbol{h}_{(0)}}(a^i) \tag{10}$$

$$\boldsymbol{m}^{ij}_{(s)} = \varphi_m(\boldsymbol{h}^i_{(s-1)}, \boldsymbol{h}^j_{(s-1)}, \boldsymbol{l}, \text{SinusoidalEmbedding}(f^j - f^i)), \tag{11}$$

$$\boldsymbol{m}^i_{(s)} = \sum_{j=1}^{N} \boldsymbol{m}^{ij}_{(s)}, \tag{12}$$

$$\boldsymbol{h}^i_{(s)} = \boldsymbol{h}^i_{(s-1)} + \varphi_h(\boldsymbol{h}^i_{(s-1)}, \boldsymbol{m}^i_{(s)}), \tag{13}$$

$$\dot{f}^i = \varphi_{\dot{f}}\left(\boldsymbol{h}^i_{(\max s)}\right) \tag{14}$$

$$\dot{l} = \varphi_{\dot{l}}\left(\frac{1}{n}\sum_{i=1}^{n} \boldsymbol{h}^i_{(\max s)}\right) \tag{15}$$

where $\boldsymbol{m}^{ij}_{(s)}, \boldsymbol{m}^i_{(s)}$ represent messages at layer $s$ between nodes $i$ and $j$, $\boldsymbol{h}^j_{(s)}$ represents hidden representation of node $j$ at layer $s$; $\varphi_m, \varphi_h, \phi_{\boldsymbol{h}_{(0)}}, \varphi_{\dot{f}}, \varphi_{\dot{l}}$ represent parametric functions with all parameters noted together as $\theta$. Finally, we define

$$\text{SinusoidalEmbedding}(x) := (\sin(2\pi kx), \cos(2\pi kx))^T_{k=0,\ldots,n_{freq}}, \tag{16}$$

where $n_{freq}$ is a hyperparameter. We standardized the $l$ input to the network with z-scoring. We also standardized the outputs for predicted tangent vectors $\dot{f}, \dot{l}$. Models were trained using the AdamW optimizer [24].

## C  Hyperparameters

We used $N_{tr} = 3.3 \times 10^6$ and trained the model for 20 epochs with early stopping. To generate the $N_{tr}$ training pairs, we used temperature $\tau = 0.9$ and nucleus probability $P = 0.99$. While other values might be explored, the high computational cost of experimentation limited our exploration of these parameters. For training the RFM model, we swept over a few values of learning rates: {1e-3, 7e-4, 5e-4, 3e-4, 1e-4}. To compute the loss function, we used loss weights $\lambda_{\boldsymbol{f}} = 200$, and $\lambda_l = 1$ in the training objective (equation (8)). These values were chosen by running a grid search over $\lambda_{\boldsymbol{f}} \in \{100, 200, 300, 400\}, \lambda_l \in \{1\}$. Additional hyperparameter settings are given in table 3.

The LLM was trained for 10 epochs with a batch size of 16, and cosine annealed learning rate of 0.0005, with LoRA rank = 8 and $\alpha = 32$.

**Compute resources** We trained our LLM model on 8x 80GB A100 GPUs for roughly 1 day. We used 4-bit quantization and LoRA to optimize training. Sampling from the trained LLM required a total of $\sim 250$ A100 GPU days, that were parallelized over 300 A100 GPUs.

Each of our RFM models were trained for 2 days on a single 32GB V100 GPU. All experiments were performed on an internal GPU cluster.

Evaluations required running DFT computations that were run on a large internal CPU cluster with 5000 nodes, each equipped with a 26-core Intel Cooper Lake-SP CPU, and 64GB memory. Each DFT computation took about 1 hour of compute on a single node, and we ran nearly 50,000 such computations to evaluate all of our models.

Table 3: RFM model hyperparameters

|  | Value |
| --- | --- |
| Hidden Dimension | 512 |
| Time Embedding Dimension | 256 |
| Number of Layers | 6 |
| Activation Function | silu |
| Layer Norm | True |
| Batch Size | 256 |
| Max Epochs | 20 |
| Inference anti-annealing scale | 5 |

# D Metrics

Thermodynamic stability is a key indicator of synthesizability, and generating novel stable materials is of keen interest in material science. Stability is determined by comparing a material's energy to those of competing crystals with the same elements. Formally, stability is measured by constructing a convex hull of all competing materials from a reference set and computing the distance from this hull (called Energy above the Hull, or $E^{\text{hull}}$). Stable materials have $E^{\text{hull}} < 0$, while materials with $E^{\text{hull}} < 0.08$ eV/atom are called metastable [39] With this defintion of stability, we define our *Stability Rate* metric as the percentage of generated materials that are stable ($E^{\text{hull}} < 0$, and n-ary $\geq 2$). For our reference set of materials, we use the Materials Project database recorded by [35] in February 2023.

Following Miller et al. [26], we compute $E^{\text{hull}}$ values by running structure relaxations on the generated structures with the CHGNet model [5] followed by density functional theory (DFT)[19] calculations.

While stability rate is an important metric, it does not capture novelty. Therefore, we define a second metric, the *S.U.N. rate* which measures the percentage of generated structures which are Stable, Unique, and Novel. To determine novelty, we exclude generated structures that are similar to any structure in the training dataset. Similarity is measured using pymatgen's StructureMatcher[29] with default settings. A generated structure that is not similar to any training data structure is considered novel.

To compute uniqueness, we use StructureMatcher to do pairwise comparisons between all generated structures, and group similar structures into equivalence classes. Each group is only counted as a single unique structure for the purpose of computing the S.U.N. rate. Formally,

$$\textit{Stability Rate} := \frac{N_{\text{stable}}}{N_{\text{gen}}} \tag{17}$$

$$\textit{S.U.N. Rate} := \frac{N_{\text{S.U.N.}}}{N_{\text{gen}}} \tag{18}$$

$$\tag{19}$$

| Method | Noise Std | Integ. Steps | Validity (%) ↑ | | Coverage (%) ↑ | |
|---|---|---|---|---|---|---|
| | | | Structural | Composition | Recall | Precision |
| FlowLLM | 0 | 250 | 99.64 | 91.99 | 94.36 | 94.38 |
| | 0.01 | 250 | 99.85 | 91.99 | 94.74 | 94.50 |
| | 0.02 | 250 | 99.99 | 91.99 | 93.41 | 94.58 |
| | 0.04 | 250 | 99.97 | 91.99 | 93.24 | 94.54 |

Table 4: Proxy metrics for a FlowLLM trained with different levels of random gaussian noise added to continuous values predicted by the LLM. Added noise increases the support of the base distribution, but we do not see an appreciable difference in the metrics.

where $N_{\text{gen}}$ is the number of generated samples, $N_{\text{stable}}$ is the number of generated samples which are stable, and $N_{\text{S.U.N.}}$ is the number of generated samples which are stable, unique, and novel.

Due to the computational expense of DFT needed to compute stability and S.U.N. rates, a number of proxy metrics have been proposed by Xie et al.[47] to benchmark model performance:

1. *Structural Validity*: Percentage of structures with valid atomic arrangements, where all pairwise interatomic distances exceed 0.5 Å.
2. *Compositional Validity*: Percentage of charge-neutral crystals, as determined by the SMACT heuristic system [4].
3. *Coverage Recall & Precision*: Standard recall and precision metrics assessing the model's ability to generate structures close to those in the test dataset. Closeness is evaluated using structural and compositional fingerprints [50, 43].
4. *Wasserstein Distances of Property Distributions*: Wasserstein distances between the distributions of computed properties (density, and $N_{\text{el}}$ – the number of unique atoms) for crystal samples from the test set and generated structures.

## E    Comparison of generated structures to ground state structures

For many practical applications in chemistry, it is important to find the local energy minimum of a generated structure. This is done by performing computationally expensive structure relaxations. Thus, it is beneficial to generate structures close to their ground state. To compare how close the generated structures are to their ground state (i.e. local energy minimum), we define 4 additional metrics (shown in table 2):

1. *Match Rate:* What fraction of generated structures and corresponding ground state structures are similar (where similarity is computed using pymatgen's StructureMatcher with default settings).
2. *RMSD:* Average RMS distance between generated structures and corresponding ground state structures computed using pymatgen's StructureMatcher whenever there is a match.
3. $\Delta$-*Energy:* Difference in energy between the generated structure and ground state structure of the DFT relaxation. This measures the reduction in energy during the structure relaxation process.
4. *Num Steps:* Number of optimizer steps needed to pre-relax the generated structure using CHGNet.

## F    Adding noise to the base distribution

Table 4 shows the effect of adding noise to the base distribution. We do not see a significant impact from the added noise.

## G    Material Generation Time

We compare the time to generate 10,000 materials between FlowLLM  with FlowMM. Inference for both models was run on a machine with a 32 core Intel(R) Xeon(R) Platinum 8488C CPU, and a single 80GB A100 GPU. FlowMM used 750 integration steps, and the RFM step of FlowLLM used

250 integration steps. With this setup, the FlowMM model takes $65.1$ minutes to generate 10,000 materials, while FlowLLM takes $89.6$ minutes, which is comparable to FlowMM.

A more useful metric is the time to generate a S.U.N. material, which is computed by dividing the inference time by the number of generated S.U.N. materials. With this metric, FlowMM takes 16.14 seconds to generate S.U.N. material, while FlowLLM takes only 10.9 seconds.

