# OpenReview forum: "FlowLLM: Flow Matching for Material Generation with Large Language Models as Base Distributions"
_NeurIPS.cc/2024/Conference — NeurIPS 2024 poster_

### Official Review · Reviewer_ixaa · 2024-07-09

**Soundness:** 3
**Presentation:** 3
**Contribution:** 2
**Rating:** 6
**Confidence:** 4

**Summary:**

This paper proposes using a second-step Riemannian flow matching (RFM) model, separately trained, to improve the quality of crystal structures generated by a pre-trained large language model (LLM), closely following Gruver et al. [1].

Specifically, the authors first follow Gruver et al. to fine-tune an LLM to enable the generation of crystal structures, and then train an RFM model from scratch to improve the stability, or in other words, quality, of the generated crystals.

The motivation of this work is clear. The reason for introducing this kind of post-processing machine learning model is to address the issue that LLMs cannot deal with real values, while atom positions and lattice parameters in crystals are usually real values.

The experimental results show that introducing this second-step RFM model can improve the stability of generated crystals. However, it faces limitations, such as the inability to generate materials with specific properties, as mentioned in the limitations section. Additionally, there are other well-established pipelines to refine the generated crystal structures that may need to be compared and discussed.


References:

[1] Nate Gruver et al. “Fine-Tuned Language Models Generate Stable Inorganic Materials as Text”. In: arXiv preprint arXiv:2402.04379 (2024).

**Strengths:**

## Strengths:

1. Clear motivation: LLMs face challenges when dealing with real values, and crystal structures are usually represented by lattice parameters and atom fractional coordinates that are real values. Introducing a post-process module to refine the generated structures of LLMs is reasonable.

2. Good performance when compared with models without such second-step refinement modules. The experimental results show that introducing such a second-step RFM model will increase the overall stability of generated crystals.

**Weaknesses:**

## Weaknesses:

1. Missing discussions with other second-stage refinement strategies that are well-established. The generated crystal structures from LLMs can also be refined by machine learning force fields (MLFF), such as M3GNet, CHGNet (which the authors have used when calculating stability), or MACE-MP. For these MLFFs, one can directly use them without training a separate RFM model from scratch to refine the generated crystal structures.

2. The contribution may be a little limited. Given that there are well-established methods for the proposed issue in this paper, the contribution of introducing an RFM model to increase stability is somewhat limited.

3. The stability rate drops significantly when removing the duplicates (to SUN rate), up to more than three times (from 17.8 to 4.92). This means a majority of the generated stable crystals are similar or the same as each other.

4. Inability to generate materials with specific properties as mentioned in the limitations section. If the ability of this model is limited to generating stable crystal structures, there are other computationally cheaper models like DiffCSP that the authors have compared with for this task. Thus, it would be beneficial to add more discussions.

**Questions:**

## Questions:
Most of my concerns are listed above in the weaknesses. Addressing the weaknesses points and the following questions may impact the final score.

1. Why not just use MLFFs like CHGNet to increase the stability of generated crystals from LLMs?

2. Are a majority of the generated stable crystals similar or the same as each other? If not, why does the stability rate drop significantly when removing the duplicates (to SUN rate), up to more than three times (from 17.8 to 4.92)?

3. Why only compare with methods without this kind of second-step refinement module?

**Limitations:**

Limitations are discussed in the main paper.

---

> ### Author Rebuttal · Authors · 2024-08-06
>
> ## > “Why not just use MLFFs like CHGNet to increase the stability of generated crystals from LLMs?”, “Why only compare with methods without this kind of second-step refinement module?”
>
> We appreciate the reviewer’s thorough knowledge on the matter of refinement, but believe a clarification is necessary. All models evaluated in our paper, including the baselines, utilize a second-step refinement process involving relaxation with a machine learning force field (specifically, CHGNet) followed by DFT relaxation. Therefore, the comparison presented in the paper is fair and focuses on the effectiveness of different generative models in producing high-quality structures that lead to stable materials after refinement.
>
> The superior stability and SUN rates we obtained for FlowLLM suggest that **our flow matching based refinement provides significant benefits BEYOND those provided by second-stage refinement with an MLFF.**
>
> One explanation for this superior performance is that MLFF relaxations only find a local energy minimum close to the generated structure, while conditional generation using riemannian flow matching does not have such a limitation. Indeed, on visual inspection of the generated structures, we found many cases where the flow matching refinement produced significantly different structures than what they started with.
> ## > The contribution may be a little limited. Given that there are well-established methods for the proposed issue in this paper, the contribution of introducing an RFM model to increase stability is somewhat limited.
>
> As mentioned above, the RFM based refinement goes beyond what an MLFF provides, and our superior results compared to other models + MLFF relaxation demonstrates this. We also address novelty in the global rebuttal above.
> ## Inability to generate materials with specific properties
> The reviewer points out the limitation of FlowLLM in generating materials with specific properties. We acknowledge this limitation in the paper, and leave extending our model to property optimization for future work.
>
> However, it's crucial to recognize that the generation of stable materials itself is a significant and challenging problem in materials science. The vast majority of theoretically possible materials are unstable and therefore not synthesizable. The ability to efficiently generate a high proportion of stable materials, as demonstrated by FlowLLM, is a substantial contribution in its own right. It dramatically reduces the search space and computational cost associated with identifying promising candidates for further investigation and potential synthesis. The focus on stability serves as a crucial filter, ensuring that the generated materials have a higher likelihood of practical relevance and applicability.
>
> We also want to point out that the LLM aspect of our proposed model leaves open the possibility of conditioning to produce specific properties in a way that is unheard of for other diffusion-based approaches: asking the LLM to produce materials with those properties. Trying to generate materials with specific properties using LLM prompting is a very interesting, albeit distinct and significant *further* contribution that we leave for future work.
> ## Stability vs SUN rate
> To compute SUN rates, we remove duplicates, and also any generated structures that are close to a structure in the training or validation data.
>
> For our best model (τ=0.7, p=0.9), 1,782 of the 10,000 generated structures are stable, out of which 926 were close to training data samples. Out of the remaining 856 structures, 492 were unique structures. Therefore, the majority of the difference between stability in SUN rates is explained by the fact that our model generates many structures close to structures in the training data. This is expected behavior for a generative model that has been trained to accurately capture the distribution of the training data. Note that the most important aspect is that the SUN rate is high, not whether the non-SUN materials are in the training data or not.
>
> We believe this offers great opportunity for future work where we can use the flexibility of LLMs for complex conditioning, while retraining the ability of our network to accurately predict stable structures (Crystal Structure Prediction) due to FlowMM’s strong performance.
> ## Cheaper methods
> While methods such as FlowMM or DiffCSP may not require tuning a LLM, they also produce far fewer SUN structures. In this dimension, FlowLLM is unequivocally better as its SUN rate is ~50% higher than the runner up.
>
> You raise an important point that one should carefully consider cost tradeoffs in this process, but they are difficult to quantify.
> 1. Inference costs have been addressed in the global rebuttal. The generation time for FlowLLM is comparable to FlowMM, which is significantly cheaper than DiffCSP, as it requires  fewer integration steps.
> 2. There is no doubt that an LLM with FlowMM will have a higher training cost, but it results in much stronger performance that cannot be replicated by merely applying existing MLFF (or DFT) relaxations to generations from the LLM. The two-step procedure makes a quantitative difference in SUN materials with higher training costs, but still low inference costs compared to DiffCSP.
> 3. Finally, the majority of the cost in a material discovery pipeline lies in the DFT relaxation that is run to check if the generated material is stable or not. Since FlowLLM generates more SUN structures, and these are generally closer to their ground state structures (see “RMSD to ground truth structures” section in the rebuttal to Reviewer sJw2), FlowLLM requires much less compute for DFT relaxations than competing methods like FlowMM. On average, structures generated by FlowLLM only need 7.08 ionic steps of DFT relaxation, while those generated by FlowMM require 14.35 steps. Therefore, **FlowLLM is able to cut the DFT cost by a factor of 2.**

---

> ### Comment · Reviewer_ixaa · 2024-08-08
> **Thanks for the rebuttal**
>
> Thank you for your responses. In summary, my concerns about following points have been addressed in a proper way.
>
> 1. My concern about “why not apply post process MLFF to other baselines” has been addressed, due to the fact that when calculating SUN scores, results from all baseline methods have been further relaxed using MLFF.
>
> 2. My concern about the contribution has been addressed, since further applying a flow model as a post process refining module is indeed helpful.
>
> My questions about the drop from stability rate to SUN rate, and the ability to do property conditioned generation have been answered, and these points seem to be valid limitations of the current proposed method.
>
> Overall, this seems to be an interesting paper to be accepted. And I increased score from 5 to 6.

---

### Official Review · Reviewer_1vvS · 2024-07-12

**Soundness:** 3
**Presentation:** 3
**Contribution:** 2
**Rating:** 6
**Confidence:** 4

**Summary:**

The paper proposed the FlowLLM, a hybrid approach for material generation that combines LLMs and RFMs, effectively leveraging their complementary strengths. Namely, this method generates samples from the parametric distribution using a two-step procedure, using 1) LLM and 2) RFM. By adopting this hybrid approach, experiments demonstrated that it exhibited better generation capabilities especially in terms of SUN and stability compared to the existing baseline.

**Strengths:**

1. Since the LLM has been trained on material data, the proposed method can greatly simplify the denoising process.
2. The approach of utilizing both LLM and RFM, two generative model approaches, to complement each other’s weaknesses is intriguing.

**Weaknesses:**

1. Since the output of the LLM is used as the starting point for the RFM denoising process, the generation time is longer compared to using a simple data distribution if there is no prior sampling of a large number ($N_{tr}$) of examples from the base distribution in advance. I would appreciate a comparison of the generation times between using 1) RFM with a simple base distribution and 2) FlowLLM without prior sampling from the LLM. Additionally, since the fewer required integration steps in the structural validity-number of integration steps plot could be attributed to the benefits of RFM, it would enhance credibility to compare this with FlowMM.
2. Using the trained LLM as the base distribution for the RFM necessitates additional LLM training costs.
3. The Preliminaries section of the paper overlaps significantly with that of the FlowMM paper. The structure and content of the Preliminaries section are very similar, which raises some questions.
4. It is insufficient to verify the performance of the proposed method on only the MP-20 dataset. I believe it would be fairer to cover all the datasets addressed in other baselines.
5. This appears to simply combine the models of CrystalLLM and FlowMM, making the novelty seem somewhat weak. If there are contributions beyond this combination, please let me know.

**Questions:**

1. Are there any experiments supporting the claim that LLMs are weak with continuous type data and denoising models are weak with discrete type data?
2. In FlowMM, bit representation was applied to atom types. Is there a specific reason for not using this approach?
3. Approximately what is the rate of generating invalid crystals?
4. Could you explain the process of how $\lambda_f$ and $\lambda_s$ were determined?
5. Could you please explain in more detail about the invariance of flat tori and fractional coordinates?

**Limitations:**

Please refer to the Weaknesses section.

---

> ### Author Rebuttal · Authors · 2024-08-06
>
> ## Generation Time
> Considering the overall generation “cost” is a good point. Could you please clarify the following?
> > I would appreciate a comparison of the generation times between using 1) RFM with a simple base distribution and 2) FlowLLM without prior sampling from the LLM.
>
> We interpret this to mean: What is the time comparison between FlowLLM and FlowMM to generate a sample? If so, we have added these to the global rebuttal.
> ## Training Time of LLMs
> While we acknowledge the additional cost of training the LLM, we believe that the significant  performance improvements justify this investment. Additionally, since we fine-tune LLAMA-2 models, the actual cost of fine-tuning these LLMs is comparable to that of DiffCSP. An additional benefit of fine-tuning LLMs is that as better pre-trained LLMs become available, the fine-tuning time and generation quality of these models will keep improving.
> ## Comparing structural validity vs NFE
> We have updated the plot comparing structural validity versus number of integration steps for FlowMM and FlowLLM (see attached fig). FlowLLM converges significantly faster than FlowMM.
> ## The Preliminaries section of the paper overlaps significantly with that of the FlowMM paper.
> We have made some revisions to the Preliminaries section to reduce the overlap, and plan to make a larger revision in the final version. Here we will outline the intended format of our preliminaries section so that you can consider it:
> 1. Introduction of crystals
>
>     a. the representation in math (similar)
>
>     b. the representation in an LLM (new)
> 2. Symmetry of crystals (shorter, focusing merely on what symmetries exist)
> 3. FlowMM (one or two paragraph summary about how FlowMM works)
> 4.CrystalLLM (one or two paragraph summary about how CrystalLLM works)
>
> The format is similar to the current paper, but we will reduce emphasis on the sections that build up flow matching for materials and instead focus on summarizing the contributions from FlowMM and CrystalLLM. We will expand the representation section so the reader understands exactly what goes into the algorithm.
> We will use the extra space to showcase new results requested by reviewers (e.g. energy/rmsd from predicted to ground state, rejection rates, more datasets) and motivate the ways in which LLMs and Flow Matching methods synergize well.
> ## More datasets
> Including other datasets is certainly a good idea, but it is unfortunately not possible to do so within the rebuttal period given the time constraints.
> As a reminder, typically used datasets are Perov, Carbon and MPTS-52. Perov and Carbon are not stable structures and therefore not relevant to unconditional generation. MPTS-52 is indeed interesting due to its chronological data split and we will focus on that for the final version of the paper.
> We want to emphasize that CrystalLLM was only tested on MP-20, and FlowMM and DiffCSP were only tested on MP-20 for De Novo Generation, which is the only task we focus on. While we agree that more datasets are better, there is significant precedent for using only MP-20.
> ## “LLMs are weak with continuous data and denoising models are weak with discrete data”
> *Denoising models are weak with discrete data:* This can be seen from the compositional validity metric, where denoising models significantly lag language models. Compositional validity measures the fraction of crystals that are charge-neutral (which is a function of the atom types). This implies that the denoising models are generating many structures with invalid atom type combinations.
>
> *LLMs are weak with continuous data:* Effectively training LLMs on materials data requires using low precision representations of atom positions [Gruver et al. 2024], which limits the range of structures that can be generated by LLMs compared to denoising models.
> This can be seen from the coverage recall metric, which measures the portion of the test distribution that is “covered” by generated structures. Crystal LLM obtains a lower recall than the denoising models (Table 1).
> ## Bit representation for atom types from FlowMM
> Our motivation was based on the following observations: 1) the atom types generated by the LLM are superior to those learned by the FlowMM using bit representations; 2) the FlowMM excelled at generating structures conditioned on the atom types (“CSP” task), significantly outperforming prior methods.
> These observations suggested to us that we should fix the atom types generated by the LLM and only use FlowMM for updating the atom positions and lattice parameters (similar to the CSP task). This allowed us to leverage the LLM’s strong comprehension of discrete data types AND FlowMM’s strong “conditional” generation capabilities.
> It would, however, be interesting to try updating the atom types during denoising in future work.
> ## How were 𝜆𝑓 and 𝜆𝑠 obtained?
> We fixed $\lambda_l = 1$, and swept over $\lambda_f \in $ {100, 200, 300, 400} (added to the manuscript).
> ## Invariance of flat tori and fractional coordinates
> Could you please clarify this question? There are a number of invariances that could be relevant.
>
> The idea of using fractional coordinates is to represent cartesian positions $x$ decomposed into a natural representation of the unit cell $l$ and the fractional coordinates $f$. Due to the fact that there are symmetries of the distribution of materials in cartesian space, some of those symmetries are inherited by the fractional coordinates. An important one is: translating all atoms by a fixed vector does not change the energy of a crystal, and therefore does not change its position in the thermodynamic competition for stability. That means we want to represent a distribution that is invariant to translation in fractional coordinates. We do this using our neural network for flow matching, like in FlowMM. In principle, the LLM part is not translation invariant, although the work by [Gruver, et. al. 24] shows that the LLM is approximately translation invariant.

---

> > ### Author Response · Authors · 2024-08-13
> >
> > Thank you again for your valuable feedback! Since we are close to the end of the discussion period, we would greatly appreciate your feedback on our response. We would be happy to provide more details, or answer any additional questions.

---

> > > ### Comment · Reviewer_1vvS · 2024-08-13
> > > **Thanks for the rebuttal**
> > >
> > > Thanks for the detailed response and summary.
> > >
> > > As they mentioned, it would be beneficial to further emphasize why using LLM and FlowMM together creates a synergistic effect. One question I have is regarding datasets like Carbon, which involves isotopes (though they mentioned the dataset is unstable). It would be interesting to explore whether using LLM is still advantageous for such datasets.
> > >
> > > Additionally, my last question was asking for a more detailed explanation of flat tori. I wanted to know the rational behind for choosing this manifold, and that has now been clarified.
> > >
> > > I have increased my score from 5 to 6.

---

### Official Review · Reviewer_sJw2 · 2024-07-13

**Soundness:** 3
**Presentation:** 4
**Contribution:** 3
**Rating:** 6
**Confidence:** 5

**Summary:**

This paper presents a generative model for crystals that uses a LLM as a base distribution and a flow matching model to refine the 3D structure. Both the LLM and flow matching parts are mostly borrowed from prior studies. However, the authors demonstrate a significant improvement in the stability of generated materials by using LLM generated crystal as the prior for the flow matching model.

**Strengths:**

The main strength of the paper is the strong empirical result that shows FlowLLM significantly improves the percentage of stable crystals by replacing the prior distribution used in FlowMM with a LLM prior. The results are validated with DFT calculations and thus convincing.

Ablation study is conducted to isolate the contribution of LLM prior. They show that the atom type is the most important factor that contributes to higher stability.

**Weaknesses:**

Despite the significant improvement in empirical results, the method presented is a straightforward combination of 2 prior works. The method is also unprincipled in some cases. For example, the LLM prior doesn’t always generate valid crystals in the defined domain. It may also generate lattices with angles outside the 60-120 range. The authors say that they simply reject invalid samples from LLM. It would be nice to report the rejection rate.

In Table 1, the SUN rate is significantly lower than the stability rate for FlowLLM. It means either the model is generating non-novel or non-unique crystals. Can the authors also report the novel and unique rates?

The wdist for density is significantly higher than DiffCSP and FlowMM. It indicates that the density of the generated crystals is less accurate.

**Questions:**

Have the authors looked at the RMSD of the generated crystals to their equilibrium structures? The wdist for density is high for FlowLLM. It might indicate that the model is generating structures far away from equilibrium

**Limitations:**

The authors noted the key limitation of FlowLLM is that it lacks end-to-end differentiability. However, the authors didn’t discuss potential negative societal impact of the model.

---

> ### Author Rebuttal · Authors · 2024-08-06
>
> ## Novelty
> Addressed in the global rebuttal.
> ## Rejection Rate of the LLM
> Addressed in the global rebuttal.
> ## Novel and Unique Rates
> The reviewer's suggestion to report the novel and unique rates separately is well-taken. For our best model, 48% of the generated structures are stable and novel, of which 58% are also unique. Novel structures are defined as those that are not close (as determined by StructureMatcher) to any structure in the training or validation sets. We have added these novel and unique rates to the revised manuscript (section 5.3).
> Note that it is possible to obtain more diverse outputs by increasing the softmax temperature of the LLM. The softmax temperature controls the randomness of the sampling process – a higher temperature leads to more diverse and creative outputs increasing the novel and unique rates, but hurts stability rates. This can be seen from our results (Table 1) where different sampling parameters lead to different stability rates, but roughly similar SUN rates. Because we can select different trade-offs between stability, novelty, and uniqueness by tweaking this parameter, we chose to focus on the SUN rate which combines all of them.
> ## wdist for density
> The higher wdist for density in FlowLLM compared to FlowMM suggests that FlowLLM's generated structures might deviate more from the test set distribution in terms of density. However, this doesn't necessarily imply less accurate crystal structures, as FlowLLM could be learning a slightly different distribution of densities that is still physically valid. The fact that FlowLLM produces significantly more stable, unique, and novel materials than FlowMM after relaxation further supports this notion. The difference in density distribution could stem from the distinct base distributions used in each model. FlowMM employs a carefully designed base distribution for the unit cell, which has been shown to improve performance. It's possible that the LLM-learned base distribution in FlowLLM is inherently more challenging to transform to the target distribution, potentially due to multimodality or disconnectedness. Investigating the impact of different base distributions and their interaction with the flow matching process is an interesting avenue for future research.
> Given that FlowLLM produces many more SUN materials after relaxation, while maintaining a high coverage, this may not be a major concern.
> ## RMSD to ground state structure
> Checking the RMSD to the equilibrium (relaxed) structure is a nice suggestion. Using methods in `pymatgen`, this requires that the relaxed structure and the generated structure “match” according to `StructureMatcher`. In addition, we also compute the average difference in energy between the initial and relaxed structures, and the average number of steps to relax the structures. All of these were computed for the CHGNet relaxation.
> The results are as follows:
> * FlowLLM: RMSD = 0.023, Match Rate = 94.9%, Delta Energy = 0.0898 eV/atom, Num steps = 37.97
> * FlowMM: RMSD = 0.096, Match Rate = 74.3%, Delta Energy = 0.3031 eV/atom, Num steps = 191.98
>
> Here, match rate is the fraction of cases where the generated and relaxed structures match. **These results show that FlowLLM actually produces structures much closer to the ground state.**
> We mentioned in the global rebuttal that inference time depends on training, generation, and (pre-)relaxation. This result shows that FlowLLM is making big improvements by reducing the number of pre-relaxation steps to find the local minimum for an atomic arrangement. We will include this in the paper as another strong result of our method.
> ## Potential negative societal impact
> We thank the reviewer for pointing this out. We have updated the broader impact section to include the following:
>
> *“The quality of technologies, e.g. batteries, depend directly on the materials that comprise them. By upgrading the underlying materials, we could improve the technology by making it more effective, cheaper, or more biodegradable. This is a double edge sword because advanced, cheap technology can both provide strong benefits to society, but can also lead to more waste or other problems. An example case is plastic. Plastic provides untold benefits to humans in convenience and preventing the spread of disease; however, it also produces a huge amount of waste that has a significant environmental impact.*
>
> *One strong point of consideration is that the materials generated by FlowLLM only exist in a computational form. It is expensive and complicated to go from a computational representation of a material with useful predicted properties to an actual technology available to the public. For this reason, we believe that the potential negative impact from computational tools for material discovery to be quite limited.”*

---

> > ### Comment · Reviewer_sJw2 · 2024-08-11
> > **Thanks for your response**
> >
> > Thanks the authors for detailed response and additional results. They've address most of my concerns, so I'd like to increase my score from 5 to 6.
> >
> > The improved RMSD is a nice result. It would be helpful if the authors can provide additional insights on the source of improvements. Can it be attributed to the better prior or the better flow matching process?
> >
> > Although I understand that authors use CHGNet to get a fast result on RMSD, DFT result is preferred for the final version of the paper.
> >
> > I also appreciate the authors adding a paragraph about potential negative societal impact.

---

### Author Rebuttal · Authors · 2024-08-06

We thank the reviewers for their detailed reviews and insightful comments. We are proud to say that all reviewers are recommending acceptance at this time. However, we aim to address any remaining critiques to improve the paper even further!

We summarize the perspective of the reviewers as follows: *Our method, FlowLLM, represents a significant improvement in empirical results when compared to existing work such as FlowMM and CrystalLLM. These results are achieved by proposing a novel and intriguing combination of LLM and Flow Matching in a way that the two methods complement and support one another. The motivation is therefore clear: take the best of the two methods and combine them. At the same time, this combination is somehow simple and therefore may have limited novelty.*

We appreciate the acknowledgement from reviewers that our method is a significant empirical improvement from previous methods. We address the critique of limited novelty below, but want to emphasize here that simplicity and novelty are not the same thing. We believe our proposal may be simple, but one that has heretofore not been explored in the literature and is therefore novel. Also, we view the simplicity of the approach as an asset rather than a hindrance.
## RMSD to ground state
Based on suggestions from one of the reviewers, we added a new set of results to the paper (Appendix D), comparing the generated structures to their ground state. We found that **FlowLLM was generating structures much closer to the ground state** than FlowMM. We believe this to be a very strong result, and re-emphasizes the efficacy of the FlowLLM method. See RMSD to ground state structure section in rebuttal to Reviewer sJw2.
## Novelty
We agree that our method combines two existing techniques. However, we respectfully disagree with the assessment that the combination is straightforward. The novelty of our contribution lies in the innovative combination of LLMs and Flow Matching:
1. **Unique Combination of LLMs & RFM:** To the best of our knowledge, no prior work has integrated the strengths of LLMs and Flow Matching together in one model for materials generation. This unique combination allows us to leverage the strengths of both methods, resulting in a substantial improvement in the rate and cost of generating stable materials.
2. **Exploiting Flow Matching's Flexibility:** FlowMM is the only prior work that uses flow matching for generating materials. The distinctive feature of Flow Matching, its ability to map to arbitrary base distributions, sets it apart from diffusion models. We are the first to recognize and leverage this property to take the output of an LLM and transform it with Flow Matching. It also opens the possibility of future work that is only possible with the flexibility of an LLM such as prompting for certain material properties.
3. **Bridging Disparate Philosophies:** The conventional wisdom in the field often views methods with inductive biases, like FlowMM, as an exclusive alternative to scalable methods like LLMs. Our work challenges this notion by successfully integrating these two seemingly disparate philosophies. The impact of such integration may be applicable beyond material generation to other domains such as proteins or small molecules. Those domains also apply strong, domain-specific inductive biases.
The substantial improvement in performance achieved by FlowLLM, as evidenced by our empirical results and ablation study, is very novel. Furthermore, the LLM's contribution extends beyond atom type prediction, serving as a learned base distribution that significantly enhances the learned target distribution. This combination and the resulting performance gains highlight the innovative nature of our work.
## Rejection Rate of the LLM
We acknowledge the limitations of the LLM prior in generating invalid crystals or lattices with angles outside the defined range. In our experiments, we observed a rejection rate of less than 0.5% due to invalid samples from the LLM (with a softmax temperature of 0.7). Note that since we can identify and reject these invalid cases, the inputs to flow matching are always valid materials. We have included this rejection rate in the revised manuscript (section 4.2).
In future work, we can explore strategies to address these limitations, such as incorporating constraints within the LLM.
## Inference Time
To compare generation speed between FlowLLM and FlowMM, we ran inference with both methods on an A100 GPU to generate 10000 samples. The time to generate 10000 samples breaks down as follows:

`InferenceTime(FlowLLM) = generate_text_time + rfm_time(250 steps) = 89.6 min`

`InferenceTime(FlowMM)  =                                    rfm_time(750 steps) = 65.1 min`

Since what we really care about is generating novel materials, a better metric is to compute the time required to generate a SUN material, which is:

`TimePerSUN(FlowLLM) = 10.9 sec / sun material`

`TimePerSUN(FlowMM) = 16.14 sec / sun material`

These results show that the inference times for the two methods are comparable, while FlowLLM generates new SUN materials faster than FlowMM.

---

### Decision · Program_Chairs · 2024-09-25

**Decision:**

Accept (poster)

**Comment:**

The paper presents a hybrid approach that combines LLMs and conditional flow models for material design. LLMs are used to generate a noisy version of the material and flow models are used to refine the initial generation. The reviewers marked the strong empirical results as one of the main strengths of the work and found the idea of combining LLMs and Flows interesting. However, they also raised concerns regarding the novelty, computational cost, and lack of comparisons against force field-based approaches which were mostly addressed in the rebuttal. Given these we are happy to recommend the paper for acceptance.